# Evaluation of Neuroprotective Effects of Quercetin against Aflatoxin B1-Intoxicated Mice

**DOI:** 10.3390/ani10050898

**Published:** 2020-05-21

**Authors:** Enrico Gugliandolo, Alessio Filippo Peritore, Ramona D’Amico, Patrizia Licata, Rosalia Crupi

**Affiliations:** 1Department of Chemical, Biological, Pharmaceutical and Environmental Science, University of Messina, 98166 Messina, Italy; egugliandolo@unime.it (E.G.); aperitore@unime.it (A.F.P.); rdamico@unime.it (R.D.); 2Department of Veterinary Science, University of Messina, 98168 Messina, Italy; plicata@unime.it

**Keywords:** aflatoxin B1, quercetin, behavioral deficit, diet, oxidative stress

## Abstract

**Simple Summary:**

Aflatoxin B1 (AFB1) is a mycotoxin commonly present in feed, characterized by several toxic effects. AFB1 has been described as being responsible for naturally occurring animal kidney disorders. In addition, AFB1 seems to have a neurotoxical effect that leads to memory impairment behavior. AFB1 toxicity involves the induction of the oxidative stress pathway, rising lipid peroxidation, and it decreases antioxidant enzyme levels. Hence, in our research, we wanted to evaluate the potential protective effects of quercetin in AFB1-mediated toxicity in the brain and the ameliorative effect on behavioral alterations. This antioxidant effect of quercetin in the brains of AFB1-intoxicated mice is reflected in better cognitive and spatial memory capacity, as well as a better profile of anxiety and lethargy disorders. In conclusion, our study suggests that quercetin exerts a preventive role against oxidative stress by promoting antioxidative defense systems and limiting lipid peroxidation.

**Abstract:**

Aflatoxin B1 (AFB1) is a mycotoxin commonly present in feed, characterized by several toxic effects. AFB1 seems to have a neurotoxical effect that leads to memory impairment behavior. AFB1 toxicity involves the induction of the oxidative stress pathway, rising lipid peroxidation, and it decreases antioxidant enzyme levels. Hence, in our research, we wanted to evaluate the potential protective effects of quercetin 30 mg/kg in AFB1-mediated toxicity in the brain and the ameliorative effect on behavioral alterations. Oral supplementation with quercetin increased glutathione peroxidase (GSH) levels, superoxidedismutase (SOD) activity and catalase (CAT) in the brain, and it reduced lipid peroxidation in AFB1-treated mice. This antioxidant effect of quercetin in the brains of AFB1-intoxicated mice is reflected in better cognitive and spatial memory capacity, as well as a better profile of anxiety and lethargy disorders. In conclusion, our study suggests that quercetin exerts a preventive role against oxidative stress by promoting antioxidative defense systems and limiting lipid peroxidation.

## 1. Introduction

One of the main factors that can lead to the deterioration of poultry feed is fungal spoilage [1]. Several factors can provide favorable conditions for the growth of fungi, like higher ambient temperatures, bad harvesting methods and inadequate drying of cereals in the field [2]. In addition, the storage of grains for longer periods due to the seasonal supply of cereals can be another factor that worsens the deterioration [1]. All these factors can bring about an increase in the development of secondary fungal metabolites correlated with mycotoxin production [3]. Certain strains of *Aspergillus flavus* and *Aspergillus parasiticus* produce secondary metabolites called aflatoxins that have been exhibited to be toxigenic, mutagenic, carcinogenic and teratogenic to several species of animals [4,5]. Among the aflatoxins, in foods and feedstuffs, aflatoxin B1 (AFB1) represents the most toxic and prevalent form [6]. AFB1 has been detected in cereal grains, as well as in entire wheat and rye breads and in many fermented beverages made from grains, oilseeds, nut products, milk, cheese, meat, fruit juices and copious other agricultural commodities [7]. Thus, aflatoxins or toxigenic fungi contamination in feed represents a potential risk to animal health. The nutritional state of an animal may greatly influence its response to toxic substances. At this date, our knowledge of the neurotoxic effects of AFB1 is poor. A large number of studies have investigated AFB1 toxicity in both the central and peripheral nervous systems of animals after exposure during embryogenesis. It was demonstrated that prenatal exposure to AFB1, in rat offspring, can delay the development of reflex response and learning ability, as well as motor activity, locomotor coordination and exploratory behavior [8,9,10]. Brain histopathological alterations that arise after progressive exposure to AFB1 in rats may include the depletion of nerve fibers, the dilation of lateral ventricles and the constriction of less distinct zones of white and gray matter in the spinal cord [11]. In addition, in Nigeria, an area with high aflatoxin exposure, researchers detected AFB1 in 81% of autopsy brain specimens from children [12]. The ability of AFB1 to penetrate the brain may be partially attributed to its detrimental effects on the blood–brain barrier’s integrity and therefore in the possible onset of cognitive deficits [13]. Quercetin, an abundant flavonoid with antioxidant properties, which commonly presents in the diet, for example, in fruits like blueberries or vegetables such as onions, curly kale, broccoli and leeks, is a promising candidate for the prevention of adverse health effects [14,15]. In the present study, we aimed to investigate the potential preventive effects of quercetin consumption in a mouse model of AFB1-induced behavioral impairment. A previous study reported the protective effects of quercetin on hepatic damage induced by AFB1 as a consequence of stabilization in the redox state [16]. The previous study provided experimental evidence for the potential beneficial actions of quercetin-rich fruits and vegetables, such as apples, onions and grapes. In fact, in a D-galactose-induced neurotoxicity mouse model, it was shown that quercetin improved spatial location learning and memory impairment in a Morris water maze (MWM) test, which was associated with a decrease in reactive oxygen species (ROS) production and an increase in antioxidant enzyme activity [17,18]. However, there has been no study of the potential AFB1 toxicity effect of impairment behavior in adult mice and the neuroprotective action of quercetin and its active components against brain behavioral disorder triggered by AFB1 [1].

## 2. Materials and Methods

### 2.1. Animals

Balb/c mice (4 months old) were housed in standard laboratory cages in a room maintained under controlled conditions of temperature (20 ± 2 °C), humidity (50% ± 5%) and photoperiod (12 h light/dark cycle). Food was available ad libitum. The University of Messina Review Board for the care of animals approved this research (n°617\2017-PR). Animal care was in accord with Italian (DM 116192) and European Economic Community regulations (OJ of EC L 358/1 12/18/1986) for the protection of experimental animals.

The animal study was conducted according to ARRIVE (Animal Research: Reporting of In Vivo Experiments) guidelines.

### 2.2. Preparation of Aflatoxin(AFB1) Extracts

Aflatoxin (AF) standards and quercetin were both purchased from Sigma Chemical Co. (St. Louis, MO, USA). AFB1 was mixed with olive oil immediately prior to oral administration to mice.

### 2.3. Experimental Design

The mice (total number = 40) were randomly allocated into five groups of eight mice each and caged separately before administration of AFB1.

Group I: control healthy mice without any treatment (*n* = 8).Group II (vehicle (veh) + control) mice received only oral administration of olive oil (200 μL/mouse/3 days) for 45 days (*n* = 8).Group III (quercetin-only) mice received only quercetin (30 mg/kg) every 3 days during the experimental period (*n* = 8).Group IV (AFB1) mice were orally administrated with 200 μL olive oil containing 25 μg AFB1 (0.75 mg/kg body weight; 1/12th of LD50) every 3 days for 45 days (*n* = 8).Group V (AFB1 + quercetin) were treated with AFB1 as in Group IV, in combination with quercetin (30 mg/kg) every 3 days for all experimental periods. (*n* = 8).

Mice were terminally anesthetized with isoflurane (Baxter International, Rome, Italy) and sacrificed by cervical dislocation at the end of the experiments. The sample size was calculated using a priori power analysis of G*Power software for statistical testing. The administration of AFB1 and/or quercetin by oral gavage was carried out every three days for 45 days. Daily intake of quercetin has been estimated to be in the range of 2–40 mg in the human or animal diet [19,20]; based on this, we selected the dosages of quercetin following previous study [21]. During the experimental period, both body weight and liquid intake were measured regularly.

AFB1 solution and quercetin were administered by oral gavage separately, first AFB1 and afterwards quercetin, between 8:00 and 9:00 a.m. AFB1 dosage was based on a previous study [9]. The whole brain tissues were dissected, washed and homogenized using a Teflon homogenizer (Heidolph Silent Crusher M, Schwabach, Germany), and subsequently were centrifuged (4 °C, 1600 g for 15 min) to obtain supernatants for biochemical investigations.

### 2.4. Behavioral Tests

#### 2.4.1. Open-Field Test (OFT)

The open-field task is a simple assessment used to calculate activity levels, gross locomotor activity and exploration habits in rodents. The test was performed as previously described [22]. Briefly, at day 45 after first treatment with AFB1, mice were placed in the central area of a round open field (56 cm in diameter), which had its floor divided into 16 equal areas. Four central areas of the apparatus had their borders limited by the walls, while the remaining peripheral areas had no contact with the wall. The time spent in central areas, and number of crossed areas (crossing), were measured as an index of anxiety-like behavior and were recorded for 5 min. For the duration of the test, the mice response was video recorded and manually counted by operators blinded to the study.

#### 2.4.2. Forced Swim Test (FST)

The forced swim test was performed according to Sunal et al., 1994 [23]. Mice were located in individual, clear polyvinyl chloride (PVC) cylinders (45 cm tall × 20 cm diameter) containing 23–25 °C water (20 cm deep to prevent the mice tails from touching the cylinder bottom). Water was changed between subjects. The immobility time during the 5 min test was recorded. Immobility was ascribed when no further activity was observed other than that required to keep the mice heads above water. Increased immobility time is an index of depressive-like behavior.

#### 2.4.3. Elevated Plus Maze (EPM)

The EPM was performed as previously described [24]. EPM consisted of a plus-shaped platform (50 cm higher than ground level) with two open (50 × 10 cm) and two enclosed arms (50 × 10 cm). Mice were placed individually on the center of the platform and allowed to freely explore for 5 min. During the test, entries of open arms and time in open arms were counted as indicators of anxiety. 

#### 2.4.4. Morris Water Maze (MWM)

To assess both spatial learning and memory, the MWM procedure was executed as described [25,26]. The escape latency, the time it took the mice to find and stand on a platform under water, was evaluated in navigation trials (3 trials per day for 5 consecutive days), and on the last day we measured the frequency time around and within the quadrant of the platform [27].

#### 2.4.5. Novel Object Recognition (NOR)

The novel object recognition test was employed for assessing recognition memory function as previously described by Botton et al. [28]. Briefly, animals were familiarized to the open-field (OF) in the absence of the objects for 10 min/day over 2 days. During the training period, animals were located in the OF with two identical objects for 10 min. Animals were considered to be exploring an object when they were dynamically sniffing the object or if the nose was investigating the familiar object in the test period, which denotes that they recognized the object before it was shown [29].

The object recognition index was calculated with the following formula: 

Recognition index = (time spent in new object) / (time spent in the new object + time spent in the already known object).

### 2.5. Malondialdehyde (MDA) Levels

Thiobarbituric acid-reactant substances evaluation, a suitable indicator of lipid peroxidation, was determined in whole brain tissues as previously described [30].

### 2.6. Cytokines Measurement

Serum tumor necrosis factor-α (TNF-α) and interleukin (IL)-1β levels were evaluated using a colorimetric commercial ELISA kit (Calbiochem-Novabiochem Corporation, Milan, Italy).

### 2.7. Evaluation of Glutathione (GSH) Levels and Superoxide Dismutase (SOD), Catalase (CAT) Activities

The levels of reduced glutathione (GSH) were determined fluorometrically in the whole brain samples following the assay reported by Mokrasch and Teschke [31]. The evaluation of superoxide dismutase (SOD) and catalase (CAT) activities in whole brain tissues were detected as indicated in a previous study [32].

### 2.8. Statistical Evaluation

All values are expressed as mean ± standard error of the mean (SEM) of n observations. A *p*-value of less than 0.05 was considered significant. The results were analyzed by one- or two-way ANOVA, followed by a Bonferroni post-hoc test for multiple comparisons. 

## 3. Results

### 3.1. Effect of Quercetin on Body Weight and Liquid Intake on AFB1 Mice

The mice survived the experimental period until sacrifice. Body weights were augmented in all experimental groups for 45 days, and weight gain did not differ among the groups (Figure 1B). There were also no significant differences in liquid intake among the experimental groups (Figure 1C).

### 3.2. Quercetin Protective Treatment on AFB1-Induced Memory Deficit

To study the cognitive functions in AFB1-intoxicated mice, novel object recognition (NOR) and MWM behavioral experiments were performed, as show in Figure 1A. In particular, during NOR training, controls and AFB1 animals showed no significant differences in their exploration time of novel objects. Forty-five days after the first AFB1 and quercetin administration, the AFB1 animals showed a significant decrease in novel object exploration time (Figure 2A). AFB1 induced an alteration of cognitive function, while the exploration time for novel object recognition was increased in mice treated with quercetin compared to the AFB1 groups (Figure 2A). An MWM test was performed to evaluate the protective action of quercetin against memory impairment. The time spent to find the platform during training was increased in AFB1-intoxicated animals compared to the veh + control (Figure 2B). AFB1 + quercetin mice showed reduced escape latency compared to AFB1 mice (Figure 2B). In addition, the frequency and time within and around the target quadrant of the platform during the probe trial was reduced in AFB1-intoxicated animals compared to AFB1 group mice (Figure 2C). Quercetin treatment increased the frequency and time, thereby ameliorating the cognitive deficits in AFB1-intoxicated mice. Finally, significant differences were shown between the veh + control group and AFB1 + quercetin in exploration time of the NOR test and frequency in the platform quadrant of MWM, that indicated there was not full recovery of AFB1 animals after treatment. No significant difference was shown in the quercetin-only group compared to veh + control mice.

### 3.3. Quercetin Protective Action on Anxiety-Like Behavior in AFB1 Mice

In order to investigate the effects of treatment with AFB1 on lethargy and anxiety-like behavior, we tested the animals using the open-field test (OFT), the elevated plus maze (EPM) and the forced swimming test (FST). We noticed significant differences in exploratory activity in terms of the number of crossings and the time spent in the center in AFB1-intoxicated mice when compared with the veh + control group in open-field performance (Figure 3A,B). Significant differences were found in immobility time. AFB1 administration increased the immobility time of the animals on the 45th day (Figure 3E). In the EPM, we noticed a decrease both in the number of entries and in the time spent in open arms after AFB1 exposition compared to the veh + control group (Figure 3C,D). Quercetin treatment significantly reduced the anxiety-like behavior in all performed tests in terms of time in the center in OF, which increased (Figure 3A,B) or reduced immobility time in FST (Figure 3E), and also, there was an increased number of entries and time spent in open arms in EPM in AFB1 mice (Figure 3C,D). Moreover, significant differences are shown between the veh + control group and AFB1 + quercetin in FST and EPM, and little difference in OF, which indicate full recovery of AFB1 animals after treatment in terms of anxiety/lethargy-like behavior. No significant difference was shown between the veh + control group and the quercetin-only group.

### 3.4. Quercetin Effects on Cytokine Levels in AFB1 Mice

We also analyzed whether proinflammatory cytokines are implicated in AFB1-induced behavioral damage. Elevated serum levels of TNF-α (Figure 4A) and IL-1β (Figure 4B) were found in the serum of AFB1 mice compared to veh + control. Quercetin significantly reduced these proinflammatory cytokine levels in AFB1 mice (Figure 4A,B). Significant differences shown between the veh + control group and AFB1 + quercetin indicated there was not full recovery of AFB1 animals after treatment in release of cytokines, and no significant difference was shown between the quercetin-only group and the veh + control group.

### 3.5. Effects of Quercetin on Lipid Peroxidation and on Oxidative Stress in AFB1 Mice

The level of MDA, an indicator of lipid peroxidation, was determined in mouse whole brain tissues after AFB1 administration or quercetin treatment. AFB1 significantly increased MDA levels in the whole brain (Table 1). GSH, CAT and SOD levels, as indicators of cellular antioxidant defense, were determined in mouse whole brain tissues after AFB1 administration. AFB1 reduced the levels of GSH, CAT and SOD enzyme activities, while quercetin increased antioxidant action in AFB1-intoxicated mice (Table 1). No significant difference was shown between the veh + control group and the quercetin-only group.

## 4. Discussion

The ability of oral quercetin to prevent behavioral and brain oxidative stress changes occurring in mice following AFB1-mediated toxicity was investigated in the current study. AFB1, known to be the most toxic of the aflatoxins, arouses particular interest because it has frequently been found as a contaminant in many feed products and it is one of the most powerful carcinogens and natural mutagens [33]. In fact, AFB1 induces toxic, carcinogenic, mutagenic and teratogenic effects in laboratory animals [34]. In addition, a previous study in pigs demonstrated the possibility of toxicosis after AFB1 contamination of food [35]. Evidence regarding AFB1 neurotoxic action is poor, though. The current literature regarding the effects of mycotoxin on cognitive ability is limited in studies concerning exposure during embryonic development. It was reported that exposure to AFB1 during mid- and late organogenesis can cause impaired conditioned avoidance learning in post-weaning rat offspring [9]. Oxidative stress plays a key role in the development of behavioral impairment; in fact, deficits like anxiety or lethargy are also associated with concomitant inflammation and oxidative damage [36]. Several studies have shown the use of antioxidant molecules as a potential therapeutic approach to behavioral disorders [37,38]. Previous investigations showed how the depressive-like behavior associated with pronounced brain oxidative stress is correlated with enhanced glutathione transferase (GST) activity and lipid peroxidation (increased MDA level), at the same time as a depleted GSH level [39]. MDA, a product of lipid peroxidation, is commonly investigated for quantifying the level of oxidative stress [40]. It is known that lipid peroxidation provokes cytotoxic damage on a wide range of organs, especially in the brain, leading to cognitive impairment and neurodegenerative diseases [41]. Moreover, GSH level reduction may be involved in neuronal death [42], and neurobehavioral and cognitive deficits in rats [43]. Several studies in AFB1-treated animals have reported the beneficial action of natural antioxidants in reducing the serum activities of various enzymes. Supplementation with lupeol, silymarin or other plant extracts inhibited the decrease in liver enzymes induced by AFB1, an effect correlated to the antioxidant properties of these compounds [44,45,46,47]. A previous study reported that quercetin is able to pass through the blood–brain barrier in in situ models [48], which suggests that quercetin could be a potential neuroprotective approach to slow degenerative disease progression. However, a discordant finding showed that quercetin, unlike quercetin pentaacetate, seemed to be ineffective at time-dependent AFB1 inhibition of binding DNA [49]. Therefore, it was suggested that the protective effects of flavonoids on AFB1-mediated toxicity can differ according to their chemical structures. In fact, nonpolar phenolic compounds lacking free hydroxyl groups have a higher potential to detoxify AFB1-mediated toxicity than other polyphenols with several hydroxyl groups [50]. Quercetin has been reported to ameliorate behavioral and cognitive impairment in Parkinson’s disease models [51], as in chronic cerebral ischemia models [52]. Moreover, quercetin administration improves cognitive deficits in colchicine- [53], scopolamine- [54] and aluminum [55]-induced learning and memory impairment models. Free radicals not only lead to lipid peroxidation of biological membranes, but also contribute to the damage of proteins and DNA during the aging process. Hence, the purpose of this study was to investigate the effects of dietary antioxidants, such as quercetin, as a means to reverse or slow the AFB1 toxicity process. Our mice behavioral analysis in NOR, OFT, EPM, FST and MWM tasks suggested a protective effect of quercetin in terms of improving brain function during AFB1-induced neuronal toxicity. In particular, our result demonstrates that AFB1 exposure is related to a worsening in memory-related test performance, suggesting an impairment in the functionality of spatial memory and recognition as shown in the MWM and NOR tasks. These effects were effectively antagonized by treatment with quercetin, which improved spatial and recognition memory in MWM and NOR tests after AFB1 exposure in mice. Furthermore, we observed an effect of AFB1 on the increase of anxious depressive behaviors, as our results in FST, OF and NOR tasks showed. In these tasks, the administration of quercetin was able to antagonize the effect of AFB1. These effects of AFB1 could be explained on the basis of its toxicity mechanism; in fact, it is widely recognized that the brain is particularly sensitive to oxidative stress and is not particularly equipped with antioxidant defenses [56]. Thus, in this study, we decided to determine the “brain oxidative stress status” to evaluate the induction of a state of oxidative stress induced by AFB1 in the brain tissue, and the potential protective effect of quercetin. In fact, our results show how prolonged exposure to AFB1 is related to a significant increase in oxidative stress; in particular, the brain tissue of mice exposed to AFB1 (750 μg/kg orally every three days for 46 days) showed a significant increase in lipid peroxidation and also an impairment in those endogenous systems responsible for endogenous antioxidant defense such as GSH, SOD and CAT. Our results are therefore in line with other studies that have also demonstrated the induction of oxidative stress in the brains and livers of rodents that were subjected to different schemes of AFB1 administration [56,57]. GSH, catalyzed by GST and conjugated with AFB1, is considered the most important detoxification reaction to protect both the liver and extrahepatic tissues from AFB1 toxicity [58,59]. The reduced brain GSH levels in AFB1-exposed mice, as recorded in our study, can be associated with increased ROS levels in the brains of mice treated with AFB1, as previously reported [57,60,61]. The high antioxidant power of quercetin is widely recognized [50], and our results demonstrate that quercetin treatment significantly reduced AFB1-induced oxidative stress, as demonstrated by the significant reduction of MDA marker of lipid peroxidation and by the increase in GSH, SOD, CAT levels in the brain tissue of mice exposed to AFB1 and treated with quercetin. The “condition of oxidative stress” and thus cell damage is closely related to the inflammatory response; in particular, it has been seen that the levels of some proinflammatory cytokines such as TNF-α and IL-1β in the brain play a key role in the induction of behavioral alterations, and in the induction of neuroinflammatory and therefore neurodegenerative processes [62,63,64]. Consistent with the results of behavioral tests and oxidative stress marker levels, we observed that in brain tissues of mice exposed to AFB1 there were significantly higher levels of IL-1β and TNF-α. Treatment with quercetin, compared to the group exposed only to AFB1, significantly reduced the levels of these cytokines, which can also be explained by the correlative reduction of oxidative stress, which in part could explain the effect of quercetin on the improvement of these mice in behavioral tests.

## 5. Conclusions

The current study presents, for the first time, evidence of the neuroprotective effects of quercetin against AFB1-induced toxicity in the mouse brain. Previous reports investigated the in vivo effects of quercetin on AFB1-induced neurotoxicity [21]. Our findings showed that the oral administration of quercetin during chronic AFB1 exposure prevented memory impairment, such as the anxiety-like behavior induced by mycotoxin in adult mice. Moreover, this protective effect of quercetin against AFB1 toxicity seemed to be related to antioxidant action in the brain, through decreased lipid peroxidation and preservation of detoxification enzymes like GSH, SOD and CAT. Thanks to data obtained from this study, we can suppose that AFB1 is able to alter brain functions and highlight the capacity of quercetin as an antioxidant to counteract these detrimental effects. Moreover, our results reinforce the idea that monitoring food preservation and, consequently, mycotoxin levels are of fundamental importance when safeguarding food safety and the well-being of both animals and humans.

## Figures and Tables

**Figure 1 animals-10-00898-f001:**
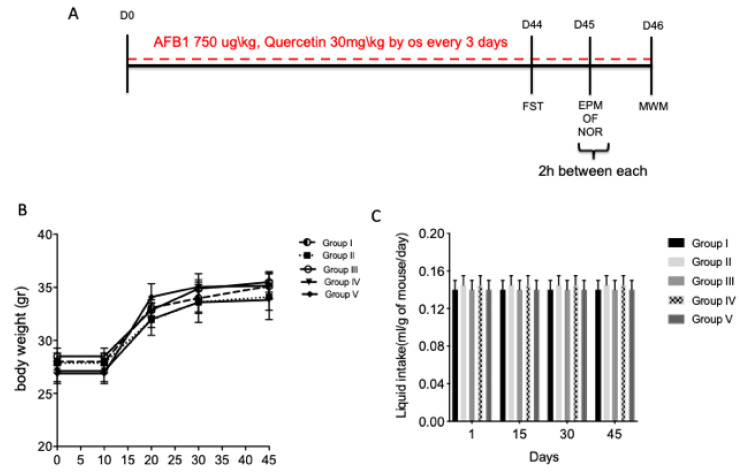
Experimental time line (**A**); (**B** and **C**). Effect of quercetin on body weight and liquid intake on aflatoxin B1 (AFB1) mice. Group I (control); Group II (veh + control); Group III (quercetin-only); Group IV (AFB1); Group V (afb1 + quercetin). Forced swimming test (FST), elevated plus maze (EPM), open-field (OF), novel object recognition (NOR) and Morris water maze (MWM). The behavioral tests were conducted on different days: FST after 44 days of first treatment; EPM, OF and NOR were conducted on day 45 in the same day, three hours between each test; MWM was conducted on day 46 of the experiment.

**Figure 2 animals-10-00898-f002:**
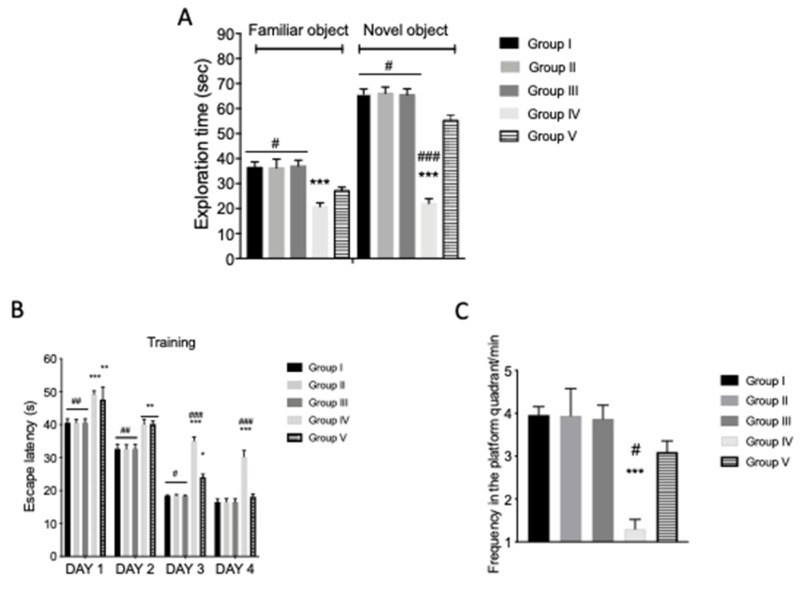
Effect of quercetin on recognition and spatial memory in AFB1 mice. Group I (control); Group II (veh + control); Group III (quercetin-only); Group IV (AFB1); Group V (afb1 + quercetin). Novel object recognition (NOR) and Morris water maze (MWM) tests were evaluated. (**A**) Exploration time in seconds; (**B**) escape latency; (**C**) frequency in the platform quadrant/min. Quercetin treatment was able to increase the exploration time for novel object recognition compared to the AFB1 group (**A**); to increase the time to finding the platform (**B**); as well as the frequency time around and within the target quadrant of the platform (**C**), compared to controls. Values are mean ± SEM of 8 mice/group. * *p* < 0.05 vs. veh + control group; # *p* < 0.05 vs. AFB1 + quercetin.

**Figure 3 animals-10-00898-f003:**
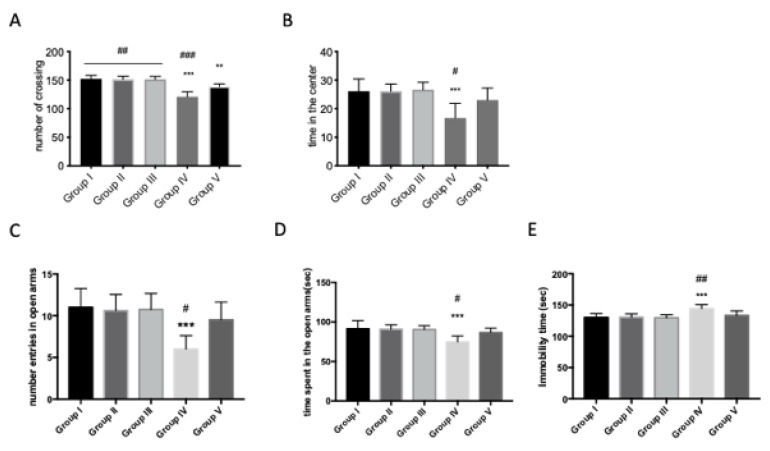
Effect of quercetin on anxiety-like behavior changes in AFB1 mice. Group I (control); Group II (veh + control); Group III (quercetin-only); Group IV (AFB1); Group V (afb1 + quercetin). Effects of 45 days of AFB1 administration on anxiety behaviors in OF. Anxiety is calculated as mean of the total time in the center, in seconds. AFB1 decreased the time spent in the center of the arena. (**A**) Regarding the number of crossings, AFB1 showed a decrease on f number of crossings (**B**). In EPM, anxiety was calculated in the number of entries in open arms (**C**); AFB1 reduced the number of entries in open arms, as well as the time spent in open arms (**D**). Quercetin showed an increase of number of entries and time spent in open arms (**C,D**). In the FST, chronic AFB1 administration had an effect on the immobility time; quercetin reduced the immobility time compared to AFB1 mice (**E**). Values are mean ± SEM of 8 mice/group. * *p* < 0.05 vs. veh control group; # *p* < 0.05 vs. AFB1 + quercetin.

**Figure 4 animals-10-00898-f004:**
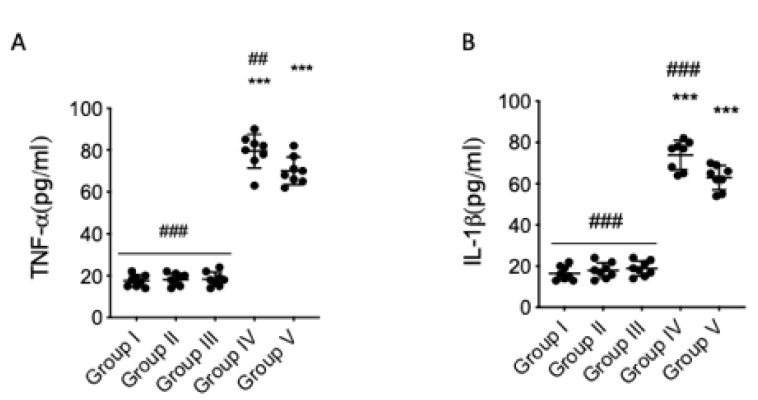
Effect of quercetin on serum cytokine in AFB1 mice. Group I (control); Group II (veh + control); Group III (quercetin-only); Group IV (AFB1); Group V (afb1 + quercetin). Serum tumor necrosis factor-α (TNF-α) (**A**) and IL-1β (**B**) proinflammatory cytokine levels were detected. Quercetin was able to reduce proinflammatory cytokines in AFB1 mice. The results are expressed as means ± SEM of 8 animals for each group. * *p* < 0.05 vs. veh control; # *p* < 0.05 vs. AFB1 + quercetin.

**Table 1 animals-10-00898-t001:** Levels of enzymic antioxidants in the brain of control and experimental animals.

Group	SOD	GSH	MDA	CAT
I	2.30 ± 0.13	27.3 ± 0.58	4.07 ± 0.15	7.75 ± 0.24 ^#^
II	2.24 ± 0.08	27.88 ± 1. 03	4.20 ± 0.35	7.36 ± 0.40 ^#^
III	2.24 ± 0.15	27.41 ± 0.69	4.34 ± 0.39	7.40 ± 0.51 ^#^
IV	1 ± 0.12 ***^,##^	20 ± 1.16 ***^,#^	6.1 ± 0.22 ***^,#^	4.2 ± 0.21 ***^,#^
V	1.70 ± 0.08	24 ± 0.94 *	5.09 ± 0.11	5.43 ± 0.16

SOD, superoxide dismutase; GSH, glutathione peroxidase; MDA, malondialdehyde; CAT, catalase. Effect of quercetin on MDA, GSH, SOD and CAT on AFB1 mice. Group I (control); Group II (veh + control); Group III (quercetin-only); Group IV (AFB1); Group V (AFB1 + quercetin). Lipid peroxidation (MDA), GSH, SOD and CAT levels were detected. Quercetin was able to reduce oxidative stress and lipid peroxidation in AFB1 mice. The results are expressed as means ± SEM of 8 animals for each group. * *p* < 0.05 vs. veh + control group; ** *p* < 0.01vs veh + control group; *** *p* < 0.001 vs. veh + control group; # *p* < 0.05 vs. AFB1 + quercetin; ## *p* < 0.01 AFB1 + quercetin; ### *p* < 0.001 vs. AFB1 + quercetin.

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
