# Peer review of "Evaluation of Neuroprotective Effects of Quercetin against Aflatoxin B1-Intoxicated Mice"

_animals, 2020, doi:10.3390/ani10050898_

Round 1

Reviewer 1 Report

In summary, the authors assessed the effects of Quercetin, a flavonoid commonly found in a variety of fruits and vegetables, on preventing or mitigating the adverse effects of AFB1 on behavioral functions and oxidative activity in mice. Overall the experiment seems sound and has useful health implications, though more details and clarity as discussed below could be warranted to improve the overall quality of the paper.

In addition, the article may require some more advanced editing for translation into English as well as an overall editorial review. There were definitely typos throughout the document and many awkwardly worded statements.  

Lines 37

“One of the main factor that can deteriorates the poultry feed’s hygiene is Fungus spoilage”

Can potentially be better stated as

“One of the main factors that can lead to deterioration of poultry feed’s hygiene is fungus spoilage”

Lines 51-53

“Literature data showed high levels of radioactivity in liver and small amounts in most other tissues of the body, including brain, 15 min to 0.5 h post-injection of AFB1 in rat and monkey”

Can potentially be better stated as

“Data from the literature shows high levels of radioactivity in liver and small amounts in most other tissues of the body, including brain, 15 min to 0.5 h post-injection of AFB1 in rat and monkey”

Also, this statement regarding toxicokinetics seems oddly placed, it does not seem to have anything to do with the previous statements. Further, what about toxicokinetics from an oral administration? That would be more applicable to this study though if that data does not exist, please note that.  

Lines 60-61

“Histopathological and neurobiochemical alterations have been associated with AFB1 exposure of rodents during adult life. However, the above clinical symptoms have not been experimentally validated.”

Please cite a source and/or if there is no source then the statements can be deleted

Lines 60-61

“In addition, in areas with high aflatoxin exposure researchers detected AFB1 in 81% of autopsy brain specimens from children living”

What specific geographic areas? Additionally “from children living” is an awkward sentence ending.

Lines 72-73

“Previous study reported the suppressive effects of Quercetin on hepatic damage induced by AFB1 and those were correlated with its antioxidant activity”

Unclear statement. Are there two studies? Also, suppressive effect on hepatic damage is an unclear mechanism, did it prevent damage or did it repair damage induced by AFB1?

Lines 75-78

“Previously study provided experimental evidences for the potential beneficial actions of Quercetin-rich fruits and vegetables, such as apples, onions, and grapes for slowing the aging effects on the brain and at the same time ameliorated the spontaneous behavior and cognitive performance through enhancement of brain inherent antioxidant capacity”

This sentence needs an editorial update. Also, it could be useful to define what was considered to be aging effect, what was considered spontaneous behavior and cognitive performance and what was considered “enhancement of brain inherent antioxidant capacity”. Technical details and definitions are useful for clarity.

2.3. Experimental Design

More detail on administration would be useful. Were the animals dosed via gavage for both Quercetin and AFB1? Were these two chemicals mixed together in one solution and then administered? If two administrations were conducted which came first, AFB1 or Quercetin?

Lines 117-118

“Body weight and liquid intake were measured regularly over the experimental period.”

This data should be presented in a figure or table and discussed. All the behavior assays in the present study are easily influenced by weight and/or nutritional effects and this data is vital for interpretation.

Lines 135-136

“The number of crossed areas (crossing) as well as the number of rearing responses (animal stands on its hind legs) was recorded for 5 minutes.”

How were these response recorded, manually or with automatic photosensors?

2.4.Behavioral tests

It can potentially be useful to provide a timeline figure/table of the behavioral tasks along with the dosing regimen. This can address the order of each task and how many days post-dosing did each task take place, and how long were the periods in between the tasks. Some of these tasks are stressful (Forced swim, Morris Water Maze) and alter homeostatic processes, which can influence behavior and potentially mask treatment effects on a subsequent task thus it’s important to address.

2.8 Statistical evaluation

“All values are expressed as mean ± standard error of the mean (SEM) of N observations. A p-value of less than 0.05 was considered significant.”

If p < 0.05 was considered significant then it can be useful to just report p < 0.05 where appropriate in all figures and tables. Reporting P < 0.01 and P < 0.001 seems unnecessary as there is no “more significant” threshold. Also, for all Results including Figures and Tables it can be very useful to report if the AFB1 + Quercetin group was significantly different from the sham group or not, to evaluate if the animals made a full recovery to expected sham levels.

Results

There is little reason not to show data for the Quercetin(30mg/kg) only group (Group V). It can be important for the reader to understand the effects of Quercetin alone on all endpoints. If possible, please present and discuss the data from Group V.

Lines 201-202

“At 45 days following first AFB1 and Quercetin administration, the AFB1 animals showed meaning fully reduced interest in the novel object”

Can potentially be better stated as

“At 45 days following first AFB1 and Quercetin administration, the AFB1 animals showed a significant decrease in novel object exploration time”

Table 1 + Figure 3

“The results are expressed as means ± SEM of 8 animals for each group. *** p < 0.001 vs. sham; # p < 0.05 vs. vehivle., ## p < 0.01 vs. vehicle.”

There is no “vehicle”. There is just the shame and the AFB1 group.

Lines 281-282

“The ability of oral Quercetin to reverse behavioral and brain oxidative stress changes occurring in mice following AFB1-mediated toxicity were investigated in the current study.”

Its not likely that it reversed any event as AFB1 and Quercetin were applied simultaneously, though Quercetin may have prevented or mitigated some of the effects of AFB1. This an important point that should be addressed as quercetin may prevent AFB1 from inducing its effects for the given exposure period.

Lines 315-362

That is a very long paragraph and should be broken up for clarity.

Lines 370-373

“Data obtained from this research, confirmed that AFB1 is able to penetrate the blood-brain barrier damaging brain functions and highlight the capacity of Quercetin as antioxidant to counteract these detrimental effects.”

There was no confirmation that AFB1 penetrated the blood brain barrier, though the evidence does potentially indicate damage of brain functions. Since AFB1’s presence in the brain was not confirmed its probably just to focus on the physiological and functional endpoints.

Reviewer 2 Report

Aflatoxn B1 is a hepatic toxin in all animal species with varying susceptibility and includes humans. It also cause carcinoma of the liver over prolonged exposure.  In the summary and abstract, the authors state that aflatoxin B1 is nephotoxic which is a major error...In their discussion, they then state it is a liver toxin which is correct.  The experimental design of the study is good, but the paper is poorly written with terrible English.  I would replace the word "depression" which is a human clinical sign expressed verbally with "lethargy" which can be observed in animals.

The area of brain analyzed needs to be defined, ie. whole brain or parts.  

Reviewer 3 Report

The paper submitted by Gugliandolo et al is well designed and performed and its quality makes it worthy of being published. Only minor things have been detected that can be modified to improve the final presentation. References should be updated, as many of them are old and others more recent can be used.

The name of the fungal strains should appear in italics.

Round 2

Reviewer 1 Report

The authors have made good-faith efforts to improve the manuscript based on feedback, however a few more important points of clarity could be useful for readers to understand and appreciate the study. The authors may benefit from increased focus on the methods and results sections to ensure that enough information and data is provided.

Lines 22-24

 Abstract: Aflatoxin B1 (AFB1) is a mycotoxin commonly present in feed, characterized by several toxic effects, in particular, AFB1 is hepatotoxic. AFB1 has been described as being responsible for naturally occurring animal kidney disorders.

Please note that these first two statements have nothing to do with AFB1 and neurotoxocity, the primary focus of the manuscript. Neither hepatoxicity nor nephrotoxicity are heavily discussed in the manuscript. It may be useful to consider focusing on more general background on AFB1 for the first statements and then introduce neurotoxicity specifically as a key issue of concern.

Lines 57-61

“Histopathological and neurobiochemical alterations have been associated with AFB1 exposure of rodents during adult life. However, the above clinical symptoms have not been experimentally validated.”

Please cite a source and/or if there is no source then the statements can be deleted

We thank the referee for the suggestion, we add reference in text as suggested

There is still no reference? The previous statements only describe developmental exposure.

Lines 72-76

“In fact, in a D-galactose-induced neurotoxicity mouse model, it was shown that quercetin improved spatial location learning and memory impairment, commonly associated with aging, in a MorrisWaterMaze(MWM) test, through a decrease in ROS production and an increase in antioxidant enzyme activity”

Edited the statement for clarity. The reference to aging seemed unnecessary or unclear, and the mode of action of quercetin in improving the behavioral types is very likely not definitive but suggestive or associative.

“In fact, in a D-galactose-induced neurotoxicity mouse model, it was shown that quercetin improved spatial location learning and memory impairment in a Morris Water Maze (MWM) test, which was associated with a decrease in ROS production and an increase in antioxidant enzyme activity”

2.3.Experimental Design

Group II, and Group V say “(data not shown).

Lines 134-135

“The number of crossed areas (crossing) as well as the number of rearing responses (animal stands on its hind legs) was recorded for 5 minutes.”

How were these responses recorded, manually or with automatic photosensors?

The response video recorded and manually counted by operator blinded to the study

Please add that language to the open field method section.

Figure 1

Please define the abbreviations (EPM, NOR, OF, etc) in the Figure 1 legend. Also the red text needs to be shifted or moved so its not cut off by the timeline bars. Figure 1 also needs more text to define and describe the information presented.

Results

There is little reason not to show data for the Quercetin(30mg/kg) only group (Group V). It can be important for the reader to understand the effects of Quercetin alone on all endpoints. If possible, please present and discuss the data from Group V.

We thank the referee for the suggestion, we didn’t insert the Quercetin group date added to shame mice, because we didn’t see any difference with the shame group, also as previously reported by Choi et al. 2010 , and for a better graphical representation we not include the group V

Certainly, appreciate the graphical representation concerns but the graphs have plenty of room for more data and again it can help emphasize to the reader to understand the effects. Also, the sham group is not defined.

Group II seems like the sham group but it states “data not shown” and Group I does not seem like a sham group as they did not receive olive oil. Also considering changing “Sham” to “vehicle control” or “vehicle-only control”. Also Group II can be considered lot more valuable then Group II as a control, given the Group II animals were gavaged everyday thereby controlling for an important factor that can influence behavior. If data can be excluded entirely, it can likely be Group I.

It can be potentially easier for full reader clarity just to show all the data, and clearly define the groups. One way is just to use the Group notation (I, II, III, etc.) and then just define these groups in the figure legends for clarity. To ease statistical considerations all groups can potentially just be compared to the vehicle control and the AFB1+Quercetin group.

Please note the ARRIVE guidelines "If any animals or data were not included in the analysis, explain why" and its underlying mandate "maximising information published". There is no clear editorial or scientific rationale to not include all the date.

Lines 295-296

“The ability of oral quercetin to reverse behavioral and brain oxidative stress changes occurring in mice following AFB1-mediated toxicity was investigated in the current study.”

As discussed in the previous review, it is not likely that quercetin reversed any event, or that the study is demonstrating that occurrence as AFB1 and Quercetin were applied simultaneously The study may be showing that quercetin may have prevented or mitigated some of the effects of AFB1. This an important point that should be addressed as quercetin may prevent AFB1 from inducing its effects for the given exposure period.

Lines 319-374

 This a very long paragraph, without a clear topic, middle and concluding statement. It could be useful to break this up into more focused paragraphs.

Lines 378-379:

“Previous reports investigated the in vivo effects of quercetin on AFB1-induced hepatotoxic lesions [23].”

This is not likely to be relevant to the point or background of the paper. Discussing previous reports on neurotoxic effects, like developmental exposure could be useful.

Author Response

Reviewer 1:

The authors have made good-faith efforts to improve the manuscript based on feedback, however a few more important points of clarity could be useful for readers to understand and appreciate the study. The authors may benefit from increased focus on the methods and results sections to ensure that enough information and data is provided.

Lines 22-24

 Abstract: Aflatoxin B1 (AFB1) is a mycotoxin commonly present in feed, characterized by several toxic effects, in particular, AFB1 is hepatotoxic. AFB1 has been described as being responsible for naturally occurring animal kidney disorders.

Please note that these first two statements have nothing to do with AFB1 and neurotoxocity, the primary focus of the manuscript. Neither hepatoxicity nor nephrotoxicity are heavily discussed in the manuscript. It may be useful to consider focusing on more general background on AFB1 for the first statements and then introduce neurotoxicity specifically as a key issue of concern.

We thank the referee for the suggestion, we have modified the text as suggested focusing on neurotoxical effect of AFB1

Lines 57-61

“Histopathological and neurobiochemical alterations have been associated with AFB1 exposure of rodents during adult life. However, the above clinical symptoms have not been experimentally validated.”

Please cite a source and/or if there is no source then the statements can be deleted

We thank the referee for the suggestion, we add reference in text as suggested

There is still no reference? The previous statements only describe developmental exposure.

We thank the referee for the suggestion, we have deleted the statements as suggested.

Lines 72-76

“In fact, in a D-galactose-induced neurotoxicity mouse model, it was shown that quercetin improved spatial location learning and memory impairment, commonly associated with aging, in a MorrisWaterMaze(MWM) test, through a decrease in ROS production and an increase in antioxidant enzyme activity”

Edited the statement for clarity. The reference to aging seemed unnecessary or unclear, and the mode of action of quercetin in improving the behavioral types is very likely not definitive but suggestive or associative.

“In fact, in a D-galactose-induced neurotoxicity mouse model, it was shown that quercetin improved spatial location learning and memory impairment in a Morris Water Maze (MWM) test, which was associated with a decrease in ROS production and an increase in antioxidant enzyme activity”

We thank the referee for the suggestion, we have modified the statement as suggested.

2.3.Experimental Design

Group II, and Group V say “(data not shown).

We thank the referee for the suggestion, we add the data of veh+control and quercetin only, that now are Group II and III.

Lines 134-135

“The number of crossed areas (crossing) as well as the number of rearing responses (animal stands on its hind legs) was recorded for 5 minutes.”

How were these responses recorded, manually or with automatic photosensors?

The response video recorded and manually counted by operator blinded to the study

Please add that language to the open field method section.

We thank the referee for the suggestion, we add the language in the section of OF method.

Figure 1

Please define the abbreviations (EPM, NOR, OF, etc) in the Figure 1 legend. Also the red text needs to be shifted or moved so its not cut off by the timeline bars. Figure 1 also needs more text to define and describe the information presented.

We thank the referee for the suggestion, we modified the abbreviations in the figure 1, also the red line and text and the description.

Results

There is little reason not to show data for the Quercetin(30mg/kg) only group (Group V). It can be important for the reader to understand the effects of Quercetin alone on all endpoints. If possible, please present and discuss the data from Group V.

We thank the referee for the suggestion, we didn’t insert the Quercetin group date added to shame mice, because we didn’t see any difference with the shame group, also as previously reported by Choi et al. 2010 , and for a better graphical representation we not include the group V

Certainly, appreciate the graphical representation concerns but the graphs have plenty of room for more data and again it can help emphasize to the reader to understand the effects. Also, the sham group is not defined.

Group II seems like the sham group but it states “data not shown” and Group I does not seem like a sham group as they did not receive olive oil. Also considering changing “Sham” to “vehicle control” or “vehicle-only control”. Also Group II can be considered lot more valuable then Group II as a control, given the Group II animals were gavaged everyday thereby controlling for an important factor that can influence behavior. If data can be excluded entirely, it can likely be Group I.

It can be potentially easier for full reader clarity just to show all the data, and clearly define the groups. One way is just to use the Group notation (I, II, III, etc.) and then just define these groups in the figure legends for clarity. To ease statistical considerations all groups can potentially just be compared to the vehicle control and the AFB1+Quercetin group.

Please note the ARRIVE guidelines "If any animals or data were not included in the analysis, explain why" and its underlying mandate "maximising information published". There is no clear editorial or scientific rationale to not include all the date.

We thank the referee for the suggestion, we insert the Group II and III data for all the figures. We changed the name groups in all the figures, and changed the statistical significance compared  to the Group II, named veh+control and the group IV AFB1+Quercetin.

Group I control healthy mice without any treatment. (n=8)

Group II(veh+control) mice received only oral administration of olive oil (200 μL/mouse/3 days) for 45 days (n=8) .

Group III(quercetin only) mice received only Quercetin(30mg/kg) every 3 days during the experimental period (n=8).

Groups IV(AFB1) mice were orally administrated with 200μL olive oil containing 25μg AFB1(0.75 mg/kg body weight; 1/12th of LD50) every 3 days for 45 days(n=8).

Group V(AFB1+Quercetin) were treated with AFB1 as in group IV in combination with Quercetin (30 mg/kg) every 3 days of all experimental periods. (n=8)

Lines 295-296

“The ability of oral quercetin to reverse behavioral and brain oxidative stress changes occurring in mice following AFB1-mediated toxicity was investigated in the current study.”

As discussed in the previous review, it is not likely that quercetin reversed any event, or that the study is demonstrating that occurrence as AFB1 and Quercetin were applied simultaneously The study may be showing that quercetin may have prevented or mitigated some of the effects of AFB1. This an important point that should be addressed as quercetin may prevent AFB1 from inducing its effects for the given exposure period.

We thank the referee for the suggestion, we modified the statement with prevent and not reverse as suggested.

Lines 319-374

 This a very long paragraph, without a clear topic, middle and concluding statement. It could be useful to break this up into more focused paragraphs.

We thank the referee for the suggestion, we managed the discussion to make it more understandable to the reader

Lines 378-379:

“Previous reports investigated the in vivo effects of quercetin on AFB1-induced hepatotoxic lesions [23].”

This is not likely to be relevant to the point or background of the paper. Discussing previous reports on neurotoxic effects, like developmental exposure could be useful

We thank the referee for the suggestion, we modified the statement with neurotoxicity effect.

Reviewer 2 Report

One small change is needed on line 384 it make Thanks a pleural not a singular.

Author Response

Reviewer 2:

One small change is needed on line 384 it make Thanks a pleural not a singular.

We thank the referee for the suggestion, we modified the word in plural as suggested